# Expression of type I collagen in response to Isoniazid exposure is indirect and is facilitated by collateral induction of cytochrome P450 2E1: An in-vitro study

Suman Santra[1¤], Debasree Bishnu[1], Gopal Krishna Dhali[1], Amal Santra[1,2], Abhijit Chowdhury[1,2]*

1 Centre for Liver Research, School of Digestive & Liver Diseases, Institute of Post Graduate Medical Education & Research, Kolkata, India, 2 JCM Center for Liver Research and Innovations, Indian Institute of Liver and Digestive Sciences, Sonarpur, India

¤ Current address: Department of Surgery, Indiana University School of Medicine, Indianapolis, IN, United States of America
* achowdhury2002@yahoo.co.in

**Data Availability Statement:** All relevant data are within the manuscript and its Supporting Information files.

## Abstract

We wanted to investigate whether Isoniazid (INH) can directly stimulate activation of hepatic stellate cells (HSCs) and enhance production of collagen. Treatment of human hepatic stellate cell line LX2 with or without 5μM INH for 24 to 72 hours was performed to look into content of cytochrome P450 2E1 (CYP2E1), activity of NADPH oxidase (NOX) and intracellular oxidative stress. Protein level as well as mRNA expression of alpha smooth muscle actin (α-SMA) and collagen1A1 (COL1A1) were assessed by western blot and real time PCR. In some experiments pyrazole (PY) was pre-treated to LX2 cells to induce CYP2E1 prior to INH treatment. CYP2E1 level as well as NOX activity was gradually increased with INH treatment in LX2 cells till 72 hours. Following 72 hours of INH exposure, intracellular glutathione (GSH) level was found to be reduced compared to control (p<0.01) and showed expression of α-SMA, indicating activation of HSC. We could not found any change in collagen expression in this experimental study. Pyrazole (PY) pre-treatment to LX2 cells caused significant increase in cellular CYP2E1 content associated with increase of NOX, intracellular reactive oxygen species (ROS), and expression of α-SMA and collagen1 after INH exposure. CYP2E1 is present in insignificant amount in HSCs and INH treatment could not induce collagen expression, although altered cellular oxidant levels was observed. But in LX2 cells when CYP2E1 was over-expressed by PY, INH administration provokes oxidative stress mediated stellate cells activation along with collagen type I expression.

## Introduction

Isoniazid (INH) is broadly used in the therapy of tuberculosis. INH, when used in combination with other drugs, is often accompanied with outcome of acute as well as chronic liver

**Funding:** This study was funded by Indian Council of Medical Research, New Delhi (Grant no.: 5/4/3-8/2011-NCD-II to A.C).

**Competing interests:** The authors have declared that no competing interests exist.

disease [1–4]. INH frequently causes drug induced liver injury (DILI) not only in India but also in many developing countries where there is high prevalence of tuberculosis. High mortality in anti tubercular drug (ATD) related acute liver failure is observed and many times, the patients require liver transplantation [5]. Moreover, chronic liver injury is reported from patients treated with INH from developing as well as developed countries [2, 4]. However, the mechanism is not yet known. Within the liver, hepatocyte is the primary site for INH metabolism. During metabolism, the conversion of INH to acetylisoniazid is mediated by N-acetyltransferase 2 (NAT2), which in turn is hydrolysed to acetylhydrazine [6]. Acetylhydrazine is further oxidized by cytochrome P450 2E1 (CYP2E1) to generate potential toxic metabolites. Hepatotoxicity triggered by INH is thought to be mediated partly by the activity of CYP2E1. CYP2E1 was found to be involved in generating oxidative stress in INH induced liver injury [7]. Within the liver, CYP2E1 content is highest in hepatocytes. As a microsomal enzyme, CYP2E1 takes a pivotal role in dealing with free radicles and thus plays an important role in hepatocyte injury related to oxidative stress. Liver scarring is a common outcome of chronic liver injury [8]. During liver injury caused by many etiologies, quiescent hepatic stellate cells (HSCs) transform into fibrogenic myofibroblasts [9]. Stimulus that causes activation of HSC usually comes from damaged hepatocytes as well as from Kupffer cells and endothelial cells [10]. But in chronic liver disease related to INH treatment, we have no information whether INH directly involves in proliferation and activation of HSC resulting collagen synthesis or these effects are derived from paracrine signals of INH induced damaged hepatocytes. This could have implications in finding out the subcellular pathways which may possibly progress to biomarkers generation and to identify targets in drug development. In this paper, we examined whether INH directly cause activation of HSC and increase collagen formation using LX2 cells, an immortalized human hepatic stellate cell line.

## Materials and methods

### Cell line and culture

We used LX2 cell line, an immortalized human hepatic stellate cells for this study. LX2 cell line was established in 2005 by Xu et al [11] at Mount Sinai School of Medicine, New York, USA. The LX2 cell line was obtained from Scott L Friedman (Mount Sinai School of Medicine, New York, USA). Dulbecco's modified Eagles medium [DMEM; (D5921; Sigma-Aldrich, St. Louis, MO)] was used to culture LX2 cells at 37°C in an atmosphere of 5% $CO_2$. The medium contained 2% fetal Bovine serum [FBS (10270106; Gibco, Thermo Fisher, USA), 1mmol/L L-glutamine (G7513; Sigma-Aldrich, St. Louis, MO) and 100 IU/ml penicillin-streptomycin (P4458; Sigma-Aldrich, St. Louis, MO).

### Treatment of cells

We treated LX2 cells at passage 4 to 6 with 5 μM INH when the cells were 60% confluent. Phosphate buffered saline [PBS(I3377; Sigma-Aldrich, St. Louis, MO) (P5368; Sigma-Aldrich, St. Louis, MO)] was used for dilution of INH. LX2 cells were exposed to either INH or PBS alone for 24 to 72 hours. In some experiments, we added antioxidants such as N-Acetyl-L-Cysteine [NAC (A7250; Sigma-Aldrich, St. Louis, MO)], TEMPOL (176141;Sigma-Aldrich, St. Louis,MO) and different inhibitors like diphenyleneiodonium [DPI (D2926; Sigma Aldrich, St. Louis, MO)], a NADPH oxidase (NOX) inhibitor and chlormethiazole, a CYP2E1 inhibitor [CMZ (C1240; Sigma Aldrich, St. Louis, MO)] in the culture medium 3 hours prior to INH treatment.

## Induction of CYP2E1 in LX2 cells

To examine the influence of CYP2E1 in INH mediated HSC activation, in separate sets of experiments; we added 50 μM pyrazole [PY (P56607; Sigma-Aldrich, St. Louis, MO] in culture medium 3 hours prior to addition of INH.

## Biochemical assays

**A) GSH assay.** Prior any treatment, cells ($1x10^6$) were seeded and incubated in DMEM media overnight. Following INH treatment for different time periods, cells were washed with PBS followed by trypsinization and centrifugation at 2000 rpm for 2 minutes. 400 μl of 10% trichloroacetic acid [TCA (T6399; Sigma Aldrich, St. Louis, MO)] was added to the cell pellet so that cellular reduced glutathione (GSH) could be extracted. At 13,000 g for 1 min, the cell suspension was centrifuged to eliminate denatured proteins. The supernatant was used to determine GSH by the enzymatic method of Tietze [12]. Briefly, 250 μl of cell supernatant was incubated with 1.25 ml of 0.1 M sodium phosphate buffer containing DTNB [5,5'-dithiobis-(2-nitrobenzoic acid; D8130; Sigma Aldrich, St. Louis, MO] and then absorbance of samples was recorded at 412 nm.

**B) Lipid peroxidation analysis.** Measurement of lipid peroxidation was carried out by a previously described method [13]. Briefly, $1X10^5$ cells were plated in six well plates prior to treatment with INH. Following overnight culture, LX2 cells were treated with INH for 24 to 72 hours. After treatment, the cells were trypsinized and suspended in PBS. The suspended cells were subjected to sonication followed by centrifugation of the cell lysate at 14000 rpm for 20 minutes. Protein was determined from the supernatant using Bradford method. To 100 μl of cell supernatant, 200 μl of 10% TCA and 400 μl of 0.67% thiobarbituric acid (TBA; T5500; Sigma Aldrich, St. Louis, MO) were added in a boiling water bath for 45 min. After being cooled in an ice bath, equal volume of tertiary butanol (19460; Sigma Aldrich, St. Louis, MO) was added to the samples followed by centrifugation at 10000 rpm for 10 min. The absorbance of the resulting supernatant was read at 532 nm to determine formation of TBA reactive components.

**C) Protein content.** RIPA buffer (9806; Cell Signaling Technology, U.S.A) was used to prepare whole cell lysate for estimation of proteins along with protease inhibitors cocktail (11836153001; Roche Diagnostics, Mannheim, Germany). Centrifugation of the lysate was carried out at 14,000×g for 20min at 4˚C. Using Bradford reagent (6916; Sigma Aldrich, St. Louis, MO) the protein content of the supernatant was measured [14]. Briefly, in 1.5 ml of Bradford reagent, 5 μl of cell supernatant was mixed well and was allowed to stand for 20 minutes in dark at room temperature. The absorbance was recorded at 595 nm spectrophotometrically.

**D) CYP2E1 activity.** Evaluation of CYP2E1 activity was determined by a previously described method [15]. Cells were harvested by trypsinization and subjected to sonication. Using differential centrifugation method, microsomes were prepared. CYP2E1 activity was assessed using 100 μg microsomal protein incubated with 0.4 mM paranitrophenol [PNP (1048; Sigma Aldrich, St. Louis, MO)] and 1 mM reduced nicotinamide adenine dinucleotide phosphate [NADPH (N9660; Sigma Aldrich,St.Louis, MO)] at 37º for 60 minutes. Following incubation, 20% TCA was added and samples were placed on ice. Just before reading, 2M sodium hydroxide (NaOH) was added. The enzymatic product was assessed at 520 nm within a few minutes by a spectrophotometer and activity of CYP2E1 was calculated.

**E) Catalase activity.** Using the protocol of Beers and Sizer [16], catalase activity assay was carried out spectrophotometrically. Briefly, in 1.9ml of 0.05M sodium phosphate buffer (pH 7) containing 0.059M hydrogen peroxide ($H_2O_2$; 8.22287; Sigma Aldrich, St. Louis, MO), 50 μg

protein was added. The reaction was initiated by adding 1ml $H_2O_2$ solution. The change in absorbance was read at 240 nm for 3 minutes and the specific activity was calculated.

**F) Glutathione peroxidase (GPx) activity.**   GPx activity was measured according to the method of Paglia and Valentine [17]. In brief, the reaction mixture consists of cell supernatant containing 50 μg of protein, 1% sodium azide (S2002; Sigma Aldrich, St. Louis, MO), 0.15 M reduced GSH (G6013; Sigma Aldrich, St. Louis, MO), yeast glutathione reductase (G3664; Sigma Aldrich, St. Louis, MO), 5mM EDTA (E6758; Sigma Aldrich, St. Louis, MO) and 0.89 mM NADPH in 0.1M sodium phosphate buffer. The reaction was initiated by the addition of 0.72 mM $H_2O_2$ (8.22287; Sigma Aldrich, St. Louis, MO). The absorbance of the samples was measured at 340 nm using a spectrophotometer. The change in absorbance was recorded at 1 min interval at 340 nm for 5 minutes.

**G) NADPH oxidase (NOX) activity.**   NOX activity was carried out as previously described [18]. The assay system comprises of cell supernatant along with 250 μmol/l NADPH with or without 10 μmol/l DPI (D2926; Sigma Aldrich, St. Louis, MO), 30 min before the assay and the rate of NADPH consumption was examined by the decrease in absorbance at 340 nm for 10 min. NOX activity was calculated using the absorption extinction coefficient of the amount of NADPH consumed which was 6.22 $mM^{-1}$ $cm^{-1}$.

## DCF staining for ROS

Assessment of intracellular ROS was done with the help of fluorescent probe 2',7'-dichloro-fluorescin diacetate (DCF-DA; D6883) obtained from Sigma-Aldrich, St. Louis, MO, to the cultured LX2 cells at various hours of INH treatment using flow cytometer (BD FACS Calibur, BD Bioscience, Pharmingen, USA) using FL1 channel to detect DCF fluorescence [19]. In brief, 30 min before the end of treatment programme with or without INH, 5 μM DCF-DA was added to LX2 cells in serum free DMEM in dark for 30 min at 37˚C. Following staining, the cells were subjected to trypsinization after washing and finally resuspended in PBS. DCF fluorescence intensity was measured using flow cytometry from 10,000 cells by Cell Quest software.

## mRNA expression using quantitative real time polymerase chain reaction

From LX2 cells RNA was extracted utilizing trizol reagent (15596026; Ambion, Life technologies, USA) according to the guidelines of the manufacturer. RNA was converted to cDNA with the help of random hexamers and reverse transcriptase as per recommendation by the commercial kit (K1622; Thermo Scientific, UK). The following primers were used for Quantitative real-time Polymerase Chain Reaction (qRT-PCR): glyceraldehydes 3-phosphate dehydrogenase (GAPDH): Forward: 5'-AGGGCTGCTTTTAACTCTGGT-3'; Reverse: 5'-CCCCACT TGATTTTGGAGGGA-3'; alpha smooth muscle actin (α-SMA): Forward: 5'-CTGTTCCAGC CATCCTTCAT-3'; Reverse: 5'-CGGCTTCATCGTATTCCTGT-3'; Collagen1A1 (COL1A1): Forward: 5'-GAACATCACCTACCACTGCA-3'; Reverse: 5'-GTTGGGATGGA GGGAGTTTA-3'; transforming growth factor beta (TGF-β): Forward: 5'-GCCCTGGACACC AACTATTGC-3';Reverse:5'GCTGCACTTGCAGGAGCGCAC-3'; tissue inhibitor of matrix metalloporoteinase (TIMP-1): Forward: 5'-AGACGGCCTTCTGCAATTCC-3'; Reverse: 5'-GAAGCCCTTTTCAGAGCCTT-3'; Matrix metalloprotienase (MMP-2): Forward: 5'- TCGC CCATCATCAAGTTCCC-3'; Reverse: 5'- TCTGGGGCAGTCCAAAGAAC-3'; Matrix metalloprotienase (MMP-9): Forward:5'-GTGATTGACGACGCCTTTGC-3'; Reverse:5'GGACCA CAACTCGTCATCGT-3'. From cDNA, qRT-PCR was executed using StepOne Plus thermocycler (ABI), primer sets and SYBR® green PCR master mix (436759; Applied Biosystems,

UK) as per instructions provided by the manufacturer. GAPDH was used as a housekeeping gene for expression of genes.

## Proliferation assay

MTT [3-(4,5-Dimethylthiazol-2-Yl)-2,5-Diphenyltetrazolium Bromide] assay was used to detect proliferation of LX2 cells according to manufacturer's instructions using the Vybrant® MTT Cell Proliferation Assay Kit (V13154; Molecular probes, Life technologies, USA). In brief, in a 96 well microplate, $1 \times 10^4$ LX2 cells/well were seeded and cultured in DMEM media for 24 hours. Following 72 hours exposure to 5 μM INH in presence or absence of 50 μM PY, fresh media was added to which 10 μl of 12 mM MTT solution was added to medium in each well of the 96-well plates. The plates were incubated at 37ºC for 4 hours. To 25 μl media, 50 μl DMSO (SRL, India) was added to each well and mixed well. At 540 nm optical density was measured using a microplate reader (Molecular Devices; USA).

## Sircol collagen assay

Sircol$^{TM}$ soluble collagen assay kit (S1000; Biocolor, UK) was used to determine total soluble collagen content as per instructions provided by the manufacturer. In brief, at the end of different time periods, media from both control and experimental groups were kept for 24 hours at 4ºC. Following centrifugation, 100 μl of each supernatant was incubated with 1 ml of Sircol dye reagent and mixed for 30 minutes. Samples were again centrifuged and to the resulting pellet 1 ml of the alkali reagent was added and reading was taken at 540 nm with a spectrophotometer.

## Western blotting

Following denaturation in sample loading buffer, forty micrograms protein was separated on sodium dodecyl sulfate polyacrylamide gel electrophoresis (SDS-PAGE). Following transfer onto Polyvinylidene fluoride (PVDF) membranes (IPVH00010, Merck millipore, India), the membrane was incubated with 5% skimmed milk (5751D; Loba chemie, India) which functions as a blocking agent. The blots were further probed with the following primary antibodies: mouse monoclonal α-SMA (sc-53015; Santa Cruz Biotechnology, Santa Cruz, USA) in a dilution of 1:300, mouse monoclonal beta actin (sc-47778; Santa Cruz Biotechnology, Santa Cruz, USA) in a dilution of 1:1000, mouse monoclonal collagen 1A1 (sc-80760; Santa Cruz Biotechnology, Santa Cruz, USA) in a dilution of 1:300, mouse monoclonal NOX1 (sc-25545; Santa Cruz Biotechnology, Santa Cruz, USA) in a dilution of 1:300, mouse monoclonal NOX2 (sc-74514; Santa Cruz Biotechnology, Santa Cruz, USA) in a dilution of 1:300, rabbit polyclonal CYP2E1 (ab53945; Abcam, USA) in a dilution of 1:300 and mouse monoclonal NOX4 (sc-518092; Santa Cruz Biotechnology, Santa Cruz, USA) in a dilution of 1:300. As a secondary antibody, we used anti mouse IgG (32430; Thermo Scientific, USA) and anti rabbit IgG (32460; Thermo Scientific, USA) in a dilution of 1:1000 conjugated with Horseradish peroxidise (HRP). Chemiluminescent detection method was used to develop blots using the Super-signal west Pico plus chemiluminescent reagent (34580; Thermo Scientific, U.S.A)

## Statistical analysis

All experiments were conducted minimum five times. We presented the data as means ± standard deviation (SD). To critically analyse statistical differences between groups, Student's t test was performed. A $p$ value less than 0.05 was considered statistically significant.

## Results

### CYP2E1 and NOX activities in LX2 cells during INH exposure

CYP2E1 plays the most vital drug metabolizing enzymes including INH. CYP2E1 produces huge amount of ROS [20]. Based on a previous study, ATD induced CYP2E1 expression resulted in apoptotic death of the hepatocytes in experimental model. [7]. However, there is no information available to clarify whether CYP2E1 is induced by INH in the HSCs provoking development of cellular oxidative stress and fibrogenesis. We therefore measured CYP2E1 activity in LX2 cells. At 0 hour, CYP2E1 activity in LX2 cells was low (14.96 ±0.69 pmol/min/mg of microsomal protein) which remain unchanged during 72 hours in control experiments (without INH treatment) (Fig 1A). Addition of INH (5 μM) in the culture medium showed slow but progressive elevation of CYP2E1 activity in LX2 cells till 72 hours of culture (Fig 1A). This finding was further confirmed by western blot (Fig 1B). NOX activity in LX2 cells was elevated during INH treatment which was parallel with the increase of CYP2E1 activity (Fig 1C). NOX is composed of many protein molecules and produce ROS not only in phagocytic cells but also in non-phagocytic cells. Further, in the liver, NOX has significant involvement during fibrogenesis. Hence, expression of both phagocytic (NOX2) and non-phagocytic (NOX1 and NOX4) isoforms of NOX was estimated in LX2 cells during INH exposure. While both control and INH treated LX2 cells exhibited expression of NOX1 protein, however the difference in expression of NOX1 between control and INH treated group at different time points was found to be insignificant (Fig 1D). Expression of NOX4 protein was not found in control as well as in INH treated LX2 cells (Fig 1D). But marked expression of NOX2 protein was observed only at 72 hours of INH treatment in LX2 cells.

### Intracellular ROS in LX2 cells during INH treatment

Intracellular ROS produced during the P450 catalytic cycle contributes to oxidative stress [21]. So we examined the intracellular ROS formation in LX2 cells during INH treatment. DCF fluorescence is a surrogate marker of intracellular hydrogen peroxide ($H_2O_2$) generation. Using flow cytometer, the percentage of LX2 cells producing ROS at different hours of INH treatment was measured. At basal level, 3% of the LX2 cells were found to generate intracellular ROS. The percentage of LX2 cells producing intracellular ROS was increased with duration of INH treatment (Fig 2A). At different hours of INH exposure, we further measured intracellular ROS level in LX2 cells by fluorescence spectrophotometry. At basal level, the DCF fluorescence, was found to be $2.0 \pm 0.23$ AFU/ $10^6$ cells. The ROS content in LX2 cells after 72 hours of INH treatment was $12 \pm 1.25$ AFU/$10^6$ cells ($p < 0.05$) [Fig 2B]. Pretreatment of LX2 cells with anti-oxidants (NAC, Tempol), NOX inhibitor DPI and CYP2E1 inhibitor CMZ prior to INH treatment significantly reduced intracellular ROS formation in response to INH treatment as depicted in Fig 2C.

### GSH level and antioxidant enzymatic activity in LX2 cells during INH treatment

GSH and other cellular antioxidant enzymes like GPx and catalase directly react with intracellular ROS produced by any source. We evaluated intracellular GSH level in LX2 cells at different time points of INH treatment. Cellular GSH level was comparable between control and INH treated LX2 cell line till 48 hours [Table 1]. But at 72 hours, cellular GSH level was significantly decreased in INH treated LX2 cells ($49.28 \pm 4.40$ nmole/mg of protein) compared to control cells ($61.53 \pm 6.82$ nmole/mg of protein) p<0.01 [Table 1]. Intracellular catalase activity exhibited no significant changes between control and INH treated LX2 cells. Activity of

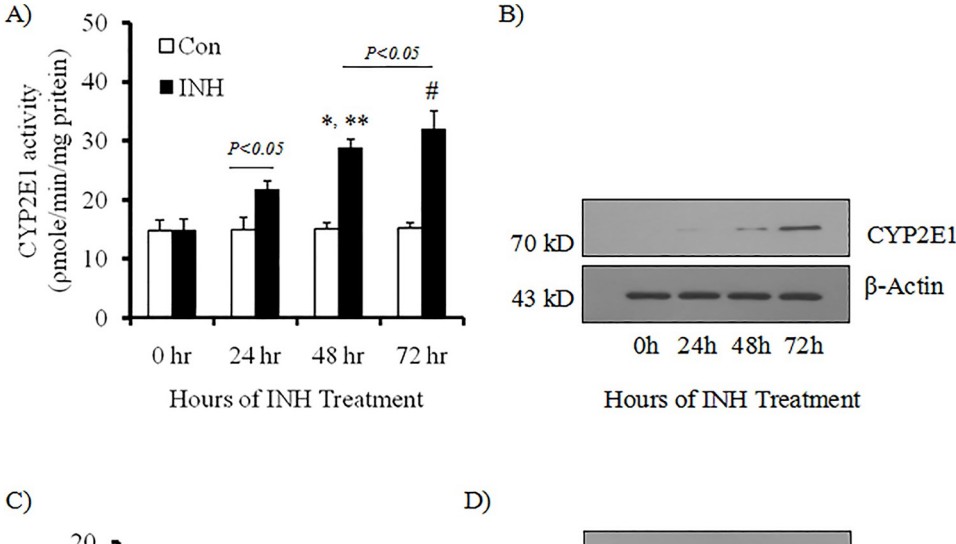

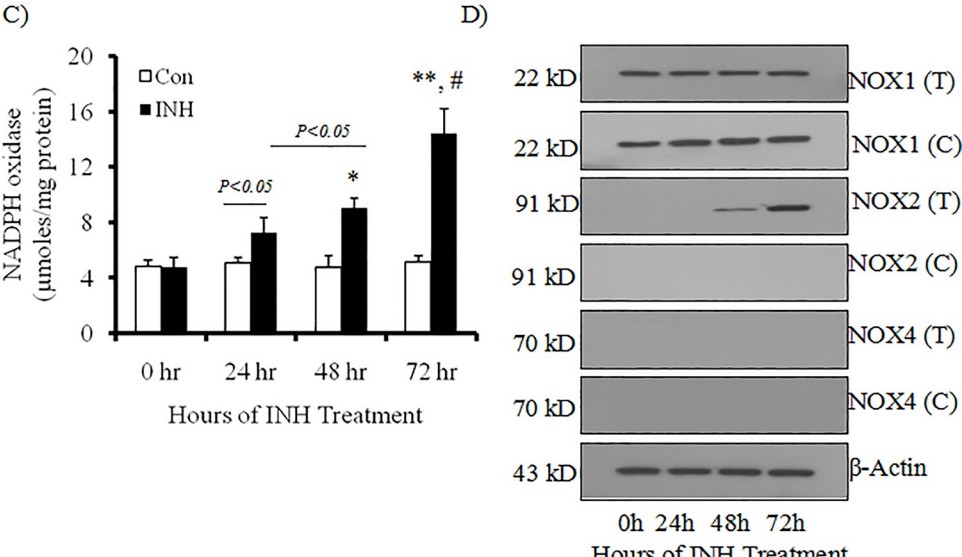

**Fig 1. Activities of CYP2E1 and NOX and their protein expression in LX2 cells at different hours of INH exposure.** LX2 cells were incubated with or without INH for 24, 48 and 72 hours (h). Further, data at 0 hour before INH treatment was also depicted in the figure. LX2 cells without INH treatment served as control. (A) Determination of CYP2E1 activity (pmol $p$-nitrocatechol/min/mg of protein) was performed. $^*$p$<$0.01 compared to 48 h control cells; $^{**}$p$<$0.01 compared to 24 h INH treated group and #p$<$0.001 compared to 72 h control cells [n = 6]. (B) Western blot of CYP2E1 protein in the cell lysates in response to INH treatment was carried out at various intervals. β-actin was used as in house control [n = 5]. (C) NOX activity [n = 6] and (D) protein expression of NOX1, NOX2 and NOX4 in the cell lysates with INH treatment (T) and without INH treatment (C) at various time periods were detected by Western blotting [n = 5]. β-actin was used as in house control. $^*$p$<$0.01 compared to 48 h control cells; $^{**}$p$<$0.001 compared to 72 h control cells and #p$<$0.01 compared to 48 h INH treated group.

cellular GPx was comparable between control and INH treatment till 48 hours as depicted in Table 1. At 72 hours, activity of GPx was significantly increased in INH treated cells (102.63 ± 3.36 U/mg protein/min) compared to control cells (64.64 ± 1.53 U/mg protein/min) p$<$0.001 [Table 1].

## Lipid peroxidation in LX2 cells during INH treatment

Lipid peroxidation products act as inducer of HSC activation. Therefore it was of interest to study the level of lipid peroxidation in LX2 cells during INH treatment. There was no

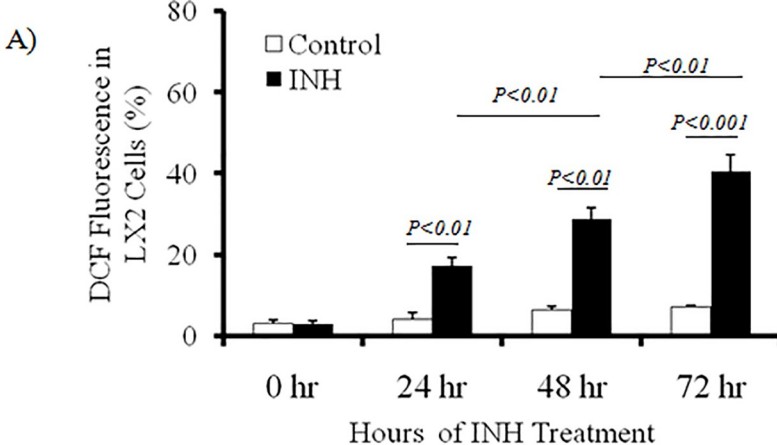

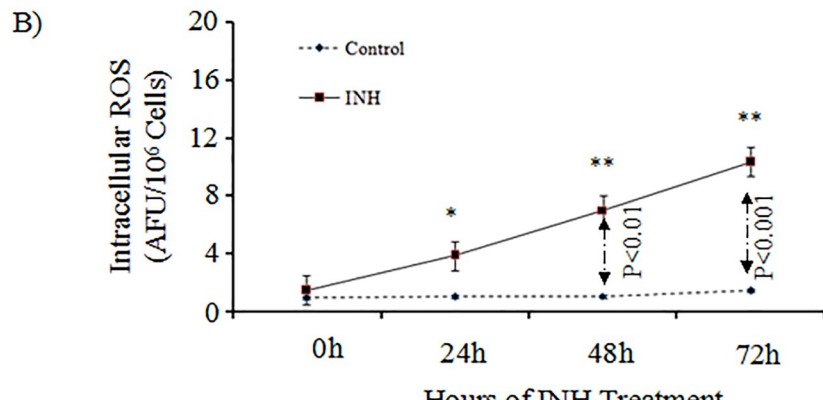

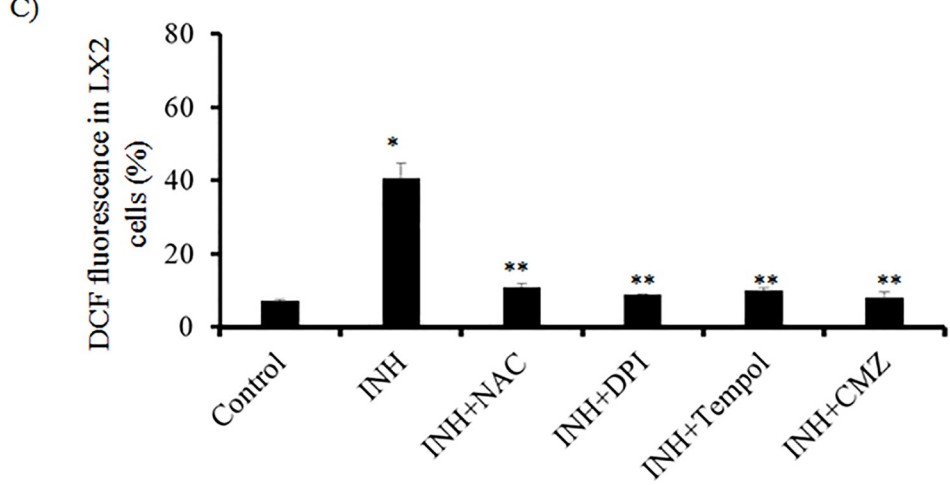

**Fig 2. Cellular ROS following INH exposure and scavenging effects of antioxidants.** Cells were treated with or without 5μM INH for 24, 48 and 72 h. Data at 0 hour before INH treatment was also depicted in the figure. For control experiments the cells were not treated with INH. (A) ROS production was evaluated by flow cytometry using 2, 7-dichlorofluorescin-diacetate (DCF-DA) and we expressed the results in terms of % of cells having DCF fluorescence [n = 5]. (B) Levels of intracellular ROS in LX2 cells were measured by fluorescence spectrophotometry using 2,7-DCF-DA as the probe. We used arbitrary units of the fluorescence intensity per $10^6$ cells to assess the level of ROS [n = 6]. *$p<0.05$ compared to 0 h; **$p<0.01$ compared to previous time points. (C) 3 hours before treatment of INH, we treated LX2 cells in presence or absence of 10μM DPI or 2mM NAC, 0.2mM Tempol and 50μM CMZ. Using flow cytometry with 2, 7-dichlorofluorescin-diacetate as the probe, determination of ROS generation was done. We

expressed the results in terms of % of cells having DCF fluorescence [n = 5]. *p<0.001 compared to control cells and **p<0.001 compared to INH treated group.

difference in the products of lipid peroxidation between INH treated and control cells till 48 hours. However, at 72 hours 1.8 fold increase of malondialdehyde (MDA) level was detected in INH treated LX2 cells compared to control cells [Table 1].

ROS and lipid peroxidation products are important mediators of stellate cell activation. Activation and perpetuation of HSCs are key steps in the development of hepatic fibrosis. We checked the expression markers of HSC activation in LX2 cells with or without INH treatment at different time intervals in both mRNA and protein level by RT-PCR and western blotting technique. After 72 hours of INH treatment to the LX2 cells, 2.1 fold increase of α-SMA mRNA expression was observed by real time PCR (Fig 3A), which was further confirmed by immunoblotting (Fig 3B). CYP2E1 inhibition by CMZ, repressed INH induced α-SMA expression in LX2 cells at 72 hours (Fig 3C). Expression of transforming growth factor beta (TGF-β), tissue inhibitors of metalloproteinase1 (TIMP1) and collagen1A1 (Col1A1) mRNA were less than 1.5 fold of the controls after 72 hours of INH treatment to LX2 cells (Fig 3A). Collagen is a major component of pathological extracellular matrix and HSCs secrete collagen upon its activation. We did not observe any collagen expression after 72 hours of INH treatment in LX2 cells neither in mRNA level nor in protein level. We also quantified total collagen released into the media using Sircol Collagen Assay kit but found no significant change in collagen level between control and INH treated LX2 cells (Fig 3D). Thus from these sets of data, we can predict that although INH can produce low level of ROS in HSCs, it does not activate HSCs to myofibroblasts and initiate collagenesis.

## Pyrazole enhances CYP2E1 activity in HSCs

Pyrazole (PY) is known to elevate CYP2E1 activity in the liver by a post transcriptional mechanism stabilizing CYP2E1 against degradation [22]. In the present study, we pretreated the LX2 cells with PY (50 μM) to induce CYP2E1 in this cell line. Addition of INH in the PY pretreated LX2 cells caused more that 2 fold increase of CYP2E1 activity at 72 hours compared to INH treatment alone (Fig 4A). Parallel to increase of CYP2E1 activity, INH also increased NOX activity in PY pretreated LX2 cells (46.48 ± 3.97 μ moles/mg protein, p<0.001 compared to only INH treatment (13.73 ± 1.27 μ moles/mg protein). NOX activity was found to be related with CYP2E1 activity (r = 0.93; p <0.001) in LX2 cells (Fig 4B). Further western blot analysis also revealed that PY pretreated INH exposed LX2 cells have higher CYP2E1 expression at 72

**Table 1. Antioxidant status and lipid peroxidation products in LX2 cell line during INH treatment.**

| Parameters | 24 Hours | | P Value | 48 Hours | | PValue | 72 Hours | | PValue |
|---|---|---|---|---|---|---|---|---|---|
| | Control | INH | | Control | INH | | Control | INH | |
| GSH (nmole/mg protein) | 60.40 ±4.47 | 61.12 ±2.35 | ns | 61.13 ± 6.08 | 59.68 ±5.23 | ns | 61.53 ± 6.82 | 49.28 ±4.40 | p<0.01 |
| GPx (μmole NADPH oxidized/min/mg protein) | 63.81 ±3.67 | 64.92 ±7.09 | ns | 68.83 ± 5.78 | 72.88 ± 6.65 | ns | 64.64 ±1.53 | 102.63 ±3.36 | p<0.001 |
| Catalase (μmole $H_2O_2$ decomposed/min/mg protein) | 5.67 ±1.45 | 5.73 ±0.67 | ns | 5.54 ± 0.91 | 5.87 ±0.71 | ns | 5.54 ±0.95 | 6.70 ±1.24 | ns |
| TBARs (nmole MDA formed/mg protein) | 0.55 ±0.03 | 0.54 ±0.21 | ns | 0.54 ± 0.06 | 0.61 ± 0.07 | ns | 0.54 ± 0.08 | 0.94 ±0.05 | p<0.001 |

GSH, reduced glutathione; GPx, glutathione peroxidase; TBARS, thiobarbituric acid reactive substances; MDA, malondialdehyde. Results are expressed as mean ± SD; n = 5.

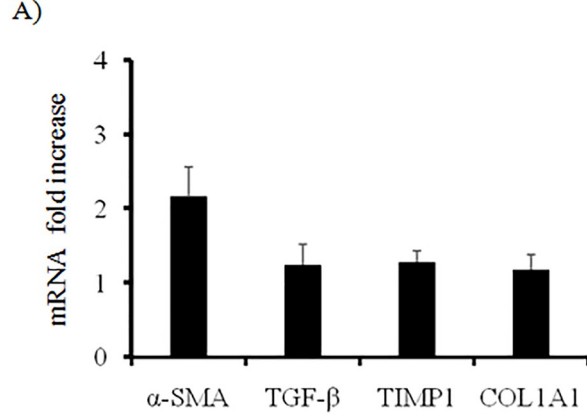

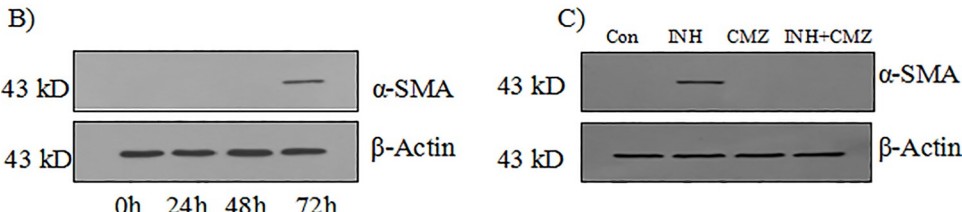

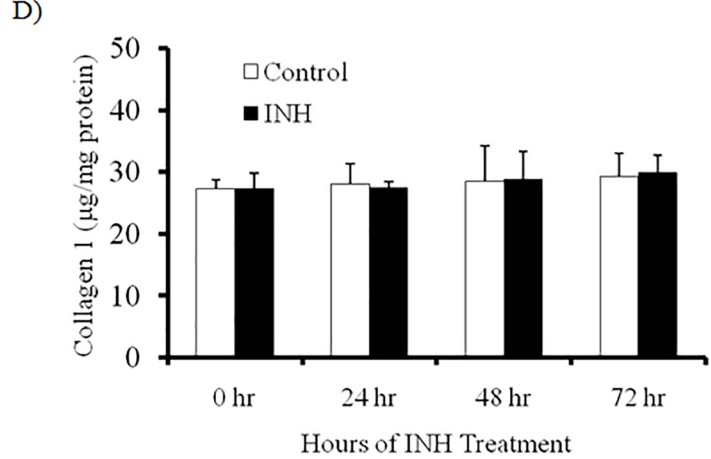

**Fig 3. The effect of INH on the activation of LX2 cells.** (A) The mRNA expression levels of Col1A1, α-SMA, TGF-β and TIMP1 were evaluated in LX2 cells after 72 hours treatment with INH by real-time PCR. RT-PCR analysis revealed that the mRNA level of α-SMA was only up regulated by 2.1 fold compared to control, while mRNA levels of Col1A1, TGF-β and TIMP1 were not up regulated (less than 1.5 fold compared to control) by INH [n = 6]. (B) Western blot analysis of α-SMA after different hours of treatment with INH in LX2 cells. Data at 0 hour before INH treatment was also depicted in the figure. β-actin was used as the house keeping gene [n = 5]. (C) Effect of CMZ on activation of LX2 cells by INH by western blot of α-SMA at 72 hours. β-actin was used as the house keeping gene [n = 5]. (D) Effect of INH on total collagen content in LX2 cells was determined by Sircol collagen assay after different hours of treatment with INH [n = 6]. Data at 0 hour before INH treatment was also depicted in the figure.

hours (Fig 4D). NOX serves as the primary source of ROS in HSCs. In HSCs, NOX1 is always present. We did not observe NOX4 expression in control LX2 cells or at different hours of

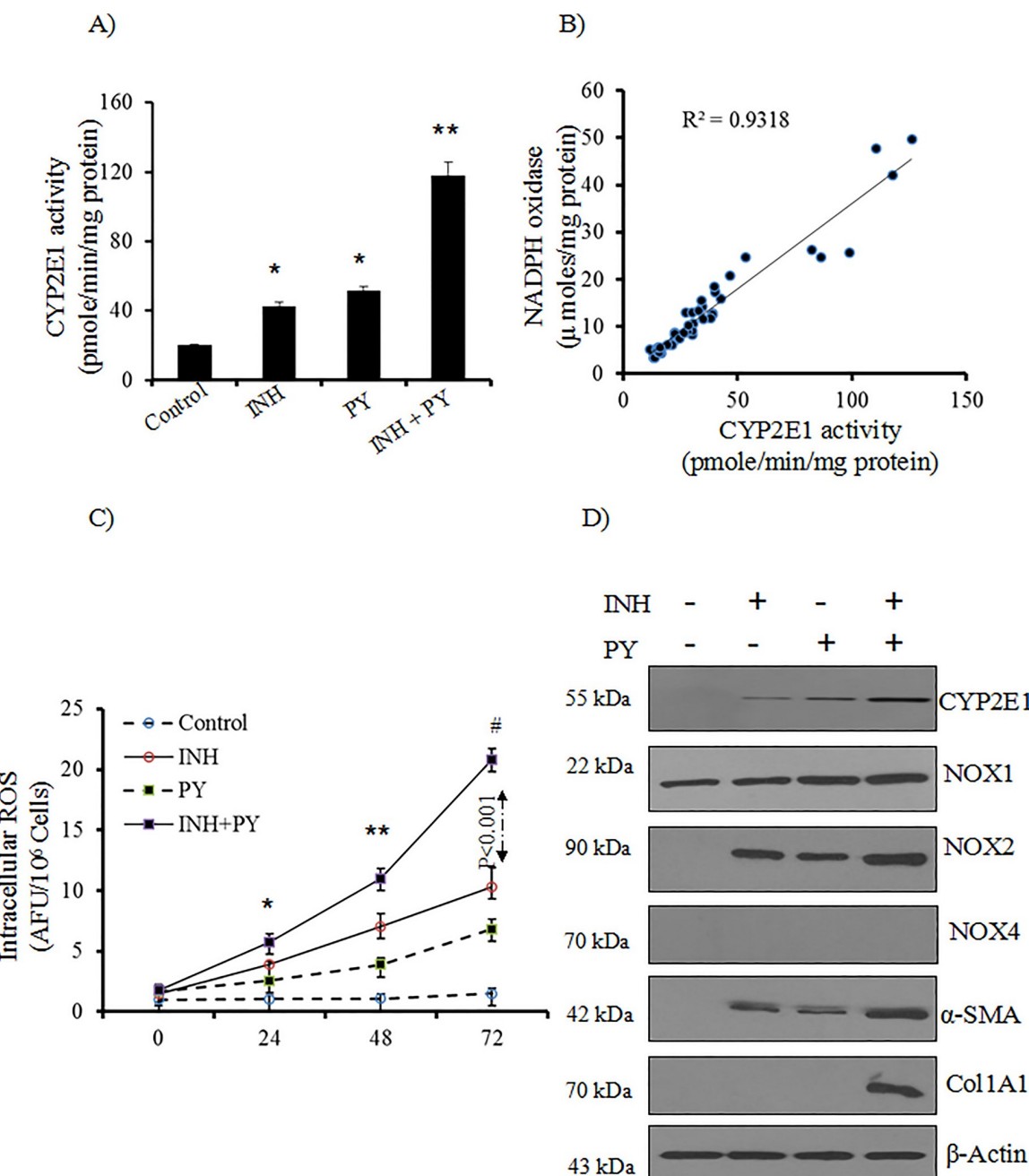

**Fig 4. The effect of pyrazole on CYP2E1 and NOX activity in LX2 cells.** Specific activity of CYP2E1 in different treatment group such as control, INH, PY, INH treatment in PY pretreated cells was determined [n = 6]. *p<0.01 compared to control cells and **p<0.001 compared to INH treated group. (B) Correlation between NOX and CYP2E1 activity in LX2 cells during INH in presence and absence of CYP2E1 inducer PY [n = 6]. (C) Levels of intracellular ROS in LX2 cells were measured by fluorescence spectrophotometry using 2,7-DCF-DA as the probe. We expressed the results in terms of arbitrary units of the fluorescence intensity per $10^6$ cells in control, INH, PY and INH treatment in PY pre-treatment groups [n = 6]. *p<0.05 compared to INH treated group; **p<0.01 compared to INH treated group and #p<0.001 compared to INH+PY treated cells of previous time points. (D) Western blot analysis of CYP2E1, α-SMA, collagen1A1, NOX1, NOX2 and NOX4 in control, INH and INH plus PY treated LX2 cells at 72 hours. PY was added in the culture 3 hours before addition of INH in the medium [n = 5]. β-actin was used as in house control.

INH treatment in PY pretreated LX2 cells (Fig 4D). By western blot analysis, NOX1 and NOX2 were found to be the predominant subunits detected in LX2 cells during the INH treatment in PY pretreated cells, however the phagocytic NOX2 was progressively increased in PY pretreated LX2 cells during INH treatment (Fig 4D).

## Pyrazole enhances INH induced oxidative stress

As depicted in Fig 4C, in the PY pretreated INH group, intracellular ROS was rapidly increased compared to INH as well as PY groups. At 72 hours, GSH level in LX2 cells remained unchanged after PY exposure ($62.33 \pm 5.83$ nmole/mg protein) compared to control cells ($61.53 \pm 6.82$ nmole/mg protein). But INH exposure in PY pretreated cells caused significant depletion of GSH ($38.28 \pm 6.43$ nmole/mg protein; $p < 0.05$) compared to INH treated group ($49.28 \pm 4.40$). Further, the level of cellular lipid peroxidation in INH exposure in PY pretreated LX2 cells was also significantly increased ($1.96 \pm 0.29$ nmole MDA formed/mg protein, $p < 0.001$) compared to only INH treated LX2 cells for 72 hours ($0.94 \pm 0.05$ nmole MDA formed/mg protein). These data indicated development of more oxidative stress in PY pretreated LX2 cells during INH treatment.

## Effect of INH, PY and INH plus PY on proliferation of LX2 cells

INH significantly increased LX2 proliferation following 72 hours of exposure as compared to control. More pronounced increase in proliferation was observed in PY pretreated INH group. Proliferation in LX2 cells was found to be markedly elevated after 72 hours of INH treatment ($0.112 \pm 0.005$; $p < 0.001$) compared to control cells ($0.062 \pm 0.004$). PY pretreated INH group exhibited further increase in cellular proliferation ($0.259 \pm 0.01$; $p < 0.001$) as compared to only PY treated group ($0.085 \pm 0.004$).

## INH activates pyrazole pretreated LX2 cells

The data of elevated CYP2E1 activity associated with increased intracellular ROS generation and evidence of increased oxidative stress in INH plus PY treated LX2 cells attempted us to evaluate activation of LX2 cells that can produce collagen 1 which is a major component of pathological extracellular matrix. So we studied α-SMA and COL1A1 mRNA level which was found to be elevated in PY pretreated LX2 cells during culture in INH treatment (Table 2).

**Table 2. Increased expression of profibrotic genes (Fold induction against control) in LX2 cells at 72 hours of INH treatment in presence and absence of CYP2E1 inducer PY.**

| Genes | Folds increase | | | |
|---|---|---|---|---|
| | INH | Pyrazole | INH + Pyrazole | P value |
| α-SMA | $2.16 \pm 0.61$ | $1.52 \pm 0.31$ | $4.42 \pm 0.15$ | ** |
| TGF-β | $1.38 \pm 0.28$ | $1.20 \pm 0.16$ | $3.16 \pm 1.06$ | * |
| TIMP-1 | $1.28 \pm 0.16$ | $1.20 \pm 0.12$ | $2.28 \pm 0.69$ | * |
| Collagen 1A1 | $1.18 \pm 0.20$ | $1.14 \pm 0.17$ | $2.82 \pm 0.49$ | ** |
| MMP-2 | $1.11 \pm 0.08$ | $1.07 \pm 0.08$ | $0.82 \pm 0.07$ | *** |
| MMP-9 | $1.07 \pm 0.08$ | $1.07 \pm 0.11$ | $0.77 \pm 0.06$ | *** |

α-SMA, alpha smooth muscle actin; TGF-β, transforming growth factor beta; TIMP-1, tissue inhibitors of metalloproteinase 1; MMP, matrix metalloproteinase. Results are expressed as mean ± SD; n = 5

*$p < 0.01$ when compared with other groups

**$p < 0.001$ when compared with other groups

***$p < 0.05$ when compared with INH group.

The expression of profibrotic genes (e.g. TGFβ1 and TIMP-1) in the PY pretreated LX2 cells were also significantly increased by INH treatment (Table 2). On the other hand, expression of both MMP-2 nd MMP-9 were significantly reduced in INH+PY treated group (Table 2). In contrast, LX2 cells when cultured without INH (control cells) or only PY treatment for 72 hours did not cause expression of α-SMA, COL1A1 and profibrotic genes. Western blotting further confirmed over expression of α-SMA and collagen 1 in PY pretreated LX2 cells by INH at 72 hours (Fig 4D).

## Discussion

We could demonstrate in the above experiments that preconditioning and collateral help by a CYP2E1 activation status modifier PY augments the ability of INH to establish a full blown profibrogenic milieu and lead to activation of collagen deposition. The sequential and graded increase of oxidative stress, HSC activation but not collagen mRNA expression by INH alone followed by a demonstration that augmentation by PY pretreatment could provide this final push is very much consonant with the pathogenesis of clinical ATD induced hepatotoxicity happening mostly with combination therapy. Fibrosis evolve more indolently than acute events and it would be important to know that INH alone is insufficient to trigger for that.

In this study we showed low level of CYP2E1 as well as intracellular ROS production following INH treatment. This low level of ROS was not able to enhance COL1A1 mRNA expression and increase collagen production. INH caused increased collagen synthesis in LX2 cells when CYP2E1 was induced in this cell line by pretreatment with PY.

CYP2E1 is involved in the activation of a number of compounds to reactive intermediates that produce tissue damage [23–25]. CYP2E1 is an active producer of ROS, which are generated from its catalytic cycle even in the absence of substrate [24, 25]. In liver, apart from hepatocytes being the primary source of CYP2E1, Kupffer cells also exhibited significant amounts of CYP2E1 [26]. But discordant reports on the CYP2E1 level in the HSC are available. According to Yamada et al, presence of CYP2E1 was maximum in the rat hepatocytes and the CYP2E1 level in rat HSC was 21% of the rat hepatocytes [27]. CYP2E1 protein expression in HSC was about 4% of the hepatocytes as observed by Oinonen et al [28]. CYP2E1 was not found to be expressed in human HSCs as reported by Casini et al [29]. CYP2E1 level was found to get elevated in liver by INH as documented from previous study [30]. In the present study, no immunoreactive CYP2E1 protein was detected in LX2 cells under basal conditions but low activity of CYP2E1 was observed. INH exposure caused slow elevation of CYP2E1 activity associated with mild intracellular ROS formation and lipid peroxidation products accumulation in LX2 cells in response to INH exposure, but the concentrations of ROS and products of lipid peroxidation are not sufficient to elicit effects on HSC activation and enhance collagen synthesis. As PY is a CYP2E1 inducer, we induced CYP2E1 level at basal condition in the LX2 cell by pretreatment with PY to examine whether INH could be bio activated rapidly in LX2 cells and increase cellular ROS production, develop oxidative stress and enhance collagen expression. We could confirm that INH can enhance collagen expression if the intracellular CYP2E1 levels in HSC remain high. High level of CYP2E1 in HSC was associated with increased cellular stress and lipid peroxidation products in response to INH exposure.

ROS are found to be one of the crucial players of stellate cell activation. Apart from the amount of ROS produced, the level of antioxidants also plays a potential role in modulating HSC activation. Oxidative stress is a phenomenal event due to excess accumulation of ROS along with severe loss of antioxidant levels. In chronic liver disease, marked depletion of antioxidant levels occur in liver. In stellate cells also depletion of antioxidants is observed which could further amplify the process of fibrogenesis [31]. This study showed evidence that in PY

pretreated LX2 cells, the increased expression of CYP2E1 by INH was associated with increased production of ROS, GSH depletion, altered antioxidants status, and expression of collagen.

Since CYP2E1 perform an essential role in DILI apart from fatty liver disorder and alcoholic liver damage, the mechanism behind regulation of CYP2E1 induction needs further investigation. A variety of exogenous substrates as well as endogenous substrates induce CYP2E1 expression that further trigger damage to hepatocytes via generation of ROS and products of lipid peroxidation. However, there is lack of experimental evidences indicating the direct involvement of the CYP2E1 inducer in the activation of HSCs.

The strength in our study is in the robustness and the novelty of the data sets, in-vitro designs pursued to address the primary question of activation of HSC and fibrogenesis during INH exposure by CYP2E1 inducer. Further in presence of CYP2E1 inducer PY, we demonstrate with precision the relevant pathophysiological changes in a sequential manner beginning with progressive development of intracellular ROS, loss of cellular antioxidants, activation of HSCs and increase collagen synthesis during INH treatment. We believe this to be the first detailed functional description of increase collagen synthesis during INH treatment in presence of CYP2E1 inducer and the connotations of the findings are fairly wide particularly in patients having alcohol abuse, intake of high fat diet and concomitant use of drugs.

In conclusion, we have been able to demonstrate that INH can lead to HSC activation directly resulting in increased collagen synthesis in the presence of CYP2E1 inducer. Our study provides initial experimental evidence to a simmering body of clinical data suggesting that drugs to be important agents in chronic hepatitis, particularly in presence of CYP2E1 inducer.

## Supporting information

**S1 Raw Images.**
(PDF)

## Acknowledgments

The authors are thankfully indebted to Professor Scott L. Friedman of Icahn School of Medicine at Mount Sinai, New York, for providing LX2 cell line as gift to Centre for Liver Research, IPGME&R, Kolkata. They acknowledge The West Bengal University of Health Science, Kolkata as the study is a part of Ph.D. thesis work of Suman Santra. The authors also acknowledge the sincere help of Pratap Pandit and Sudipta Chakraborty for their technical support.

## Author Contributions

**Conceptualization:** Amal Santra, Abhijit Chowdhury.

**Data curation:** Suman Santra.

**Formal analysis:** Suman Santra.

**Funding acquisition:** Amal Santra, Abhijit Chowdhury.

**Investigation:** Suman Santra, Debasree Bishnu.

**Methodology:** Amal Santra, Abhijit Chowdhury.

**Project administration:** Amal Santra.

**Resources:** Amal Santra.

**Software:** Suman Santra.

**Supervision:** Amal Santra.

**Validation:** Amal Santra.

**Visualization:** Suman Santra.

**Writing – original draft:** Amal Santra, Abhijit Chowdhury.

**Writing – review & editing:** Gopal Krishna Dhali.

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
