## [Decision Letter · Decision Letter 0]

10 Feb 2020

PONE-D-19-35497

Expression of type I collagen in response to Isoniazid exposure is indirect and is facilitated by collateral induction of Cytochrome P450 2E1: An in-vitro study

PLOS ONE

Dear Dr. Chowdhury,

Thank you for submitting your manuscript to PLOS ONE. After careful consideration, we feel that it has merit but does not fully meet PLOS ONE’s publication criteria as it currently stands. Therefore, we invite you to submit a revised version of the manuscript that addresses the points raised during the review process.

Your manuscript was reviewed by two experts and both of them suggested major revision. Many technical details were missing from the manuscript.

We would appreciate receiving your revised manuscript by Mar 26 2020 11:59PM. To enhance the reproducibility of your results, we recommend that if applicable you deposit your laboratory protocols in protocols.io, where a protocol can be assigned its own identifier (DOI) such that it can be cited independently in the future. For instructions see: http://journals.plos.org/plosone/s/submission-guidelines#loc-laboratory-protocols

We look forward to receiving your revised manuscript.

Kind regards,

Partha Mukhopadhyay, Ph.D.

Academic Editor

PLOS ONE

Journal Requirements:

Reviewers' comments:

Reviewer's Responses to Questions

**Comments to the Author**

1. Is the manuscript technically sound, and do the data support the conclusions?

Reviewer #1: Partly

Reviewer #2: Yes

2. Has the statistical analysis been performed appropriately and rigorously? 

Reviewer #1: Yes

Reviewer #2: Yes

3. Have the authors made all data underlying the findings in their manuscript fully available?

Reviewer #1: Yes

Reviewer #2: Yes

4. Is the manuscript presented in an intelligible fashion and written in standard English?

Reviewer #1: Yes

Reviewer #2: Yes

5. Review Comments to the Author

Reviewer #1: The authors have achieved substantially in investigating the underlying mechanism of isoniazid (INH)-dependent hepatocyte stellate cell (HSC) activation. The manuscript is structured well and proceeds smoothly with a logical flow. However, the manuscript presents certain fundamental flaws addressing which will improve the manuscript considerably.

Major

• The authors need to have every detail of the reagents and chemicals used in each experiment mentioned in the manuscript. Various sub-sections of the “materials and methods” section (cell line and culture, treatment of cells, and induction of CYP2E1) lack relevant details including catalog numbers and vendors for the reagents used in this study. I will suggest that the authors incorporate the information in the sub-sections mentioned above. Moreover, the authors will need to include the information on the respective primary antibody dilutions in the “western blotting” sub-section too.

• In the “biochemical assays” sub-section of the “materials and methods,” the authors do not elaborate on the technique used to measure the reduced intracellular glutathione (GSH), thiobarbituric acid reactive substances (TBARs), and protein content. Providing at least a brief overview of the technique is necessary, in addition to the provided references. Similarly, I would like to point to the absence of a detailed explanation of the assay technique used for measuring the activities of CYP2E1, catalase, glutathione peroxidase (GPx), and NADPH oxidase (NOX). Since determining the activities of these proteins are a crucial part of the presented data, it is of great importance that the authors explain the activity assay used for this study.

• The authors have performed the Sircol Collagen Assay (line 255) and presented the result in Figure 3C. There is, however, no mention of the assay technique in the “materials and methods” section. I want to emphasize that every experimental procedure should be documented with enough details for any researcher to reproduce it if needed. I will suggest the authors to provide the catalog number and vendor as well.

Minor

• Some of the paragraphs or portions of paragraphs are underlined within the manuscript. Is it done deliberately, what does it mean? I will suggest that the authors be consistent in their manuscript preparation.

• The authors indicate in the “statistical analysis” that all the experiments have were replicated at least five times (line 147). None of the individual figures possess any information on the number of replicates used for each experiment (Figures 1 through 4). I will suggest that the authors provide the exact number of experimental replicates used to prepare the quantification graphs in each figure.

Reviewer #2: English writing in the manuscript need to be improved.

Question 1:

In Figure 1D, the expression of NOX was detected by western blot. What are the NOX1 (T) and NOX1 (C)? The results and figure legend didn’t explain these items.

The results need to show the expression of NOX in untreated cells. After INH treatment, there is not the obvious increase in NOX1 expression between 24h and 72h. But, due to the lack of the results in untreated cells, I didn’t know the change in NOX1 expression after INH treatment compare to the untreated control.

Question 2:

In Figure 4A, the results showed the CYP2E1 activity was increased after INH, PY or INH+PY treatment. Could the authors also show the results using western blot about the expression of CYP2E1 after INH, PY or INH+PY treatment?

Question 3:

CYP2E1 is expressed mainly in hepatocytes and generates reactive oxygen species (ROS). Accumulation of reactive oxygen species (ROS) serves as a driving force for HSC activation. HSC cells expressed very low level CYP2E1. After INH 72 h treatment, the CYP2E1 expression is slightly increased in HSC cells. The increased CYP2E1 after INH treatment could not increase the collagen synthesis. In order to increase the expression of CYP2E1 in HSC cells, the author used pyrazole (PY) pre-treated HSC cells to increase the expression of CYP2E1. Since Hepatocytes can express the high level CYP2E1, Why the authors mimic the in vivo environment to use co-cultured transwell method to investigate whether INH-induced CYP2E1 expression in hepatocytes affect the HSC activation in vitro culture assay?

Question 4:

The results showed INH or PY+INH treatment can induce HSC activation. Upon activation, HSCs express a combination of MMPs and TIMPs to remodel the ECM and facilitate the progression of liver fibrosis. How about the proliferation of HSC cells and MMP expression after INH or PY+INH treatment?

Question 5:

In Figure 2C, the results showed pretreatment of LX2 cells with anti-oxidants or NOX inhibitor can reduce the intracellular ROS formation in response to INH treatment. How about the effect of CYP2E1 inhibitor in the same experiment?

Since the authors demonstrated that INH-induced CYP2E1 expression can activate HSC cells, Whether the CYP2E1 inhibitor can reverse the effect in response to INH treatment?

6. PLOS authors have the option to publish the peer review history of their article (what does this mean?). If published, this will include your full peer review and any attached files.

Reviewer #1: No

Reviewer #2: No

---

## [Author Response · Author response to Decision Letter 0]

6 May 2020

Response to Reviewers

We would like to thank the academic editor and the reviewers for critical review of our manuscript and for their thoughtful comments and constructive suggestions, which help to improve the quality of this manuscript. Point wise answer of additional requirements and the comments of the reviewers are follows: 

Response in relation of additional requirements:

“Please ensure that your manuscript meets PLOS ONE’s style requirements, including those for the naming”

Answer: We have carefully revised our manuscript as per format of PLOS ONE style. 

　　2. “ORCID ID of the corresponding author”

Answer: The ORCID ID of the corresponding author, Dr. Abhijit Chowdhury, is orcid.org/0000-0001-9198-4436.

“PLOS ONE requires the authors provide the original uncropped and unadjusted images underlying blots or gel results reported in a submission’s figures or supporting information files”

Answer: Original uncropped and unadjusted images are submitted in a separate file “S1_raw_images” as per suggestion.

　　We are really sorry for not providing all the uncropped and unadjusted images of blots for Figure 1D. Dr. Suman Santra left the department to join his Post Doctoral research program in USA and Dr. Amal Santra on superannuation of his service at IPGME&R, Kolkata, left the department and joined at JCM Center for Liver research and Innovation Kolkata. His lab at IPGME&R was shifted to JCM Center for Liver Research and innovation, Kolkata. During shifting his lab, some of the documents were misplaced and missing. This is the reason; we are unable to submit the uncropped and unadjusted images of blots of NOX1 (T) and NOX2 for figure 1D. Hope, the academic editor as well as the reviewers will sympathetically consider our case.

　　Further, in our lab, we carried out western blot development protocol using the sensitive chemiluminescent substrate (Supersignal west Pico plus chemiluminescent reagent (34580; Thermo Scientific, U.S.A) for the detection of horseradish peroxidase (HRP) exposed to X-ray film. For gel electrophoresis and western blot, we used Thermo Scientific SpectraTM Multicolor Broad Range Protein Ladder (26634) which is a prestained protein ladder. This is a ready to use ladder which can be directly loaded into gels. Since it is a prestained protein ladder, we couldnt visualize any signal for the protein marker lane after development protocol onto the exposed Xray film except for our desired protein of interest. 

“We noted that you have included the phrase “Data not shown” included in your manuscript. Unfortunately this does not meet the journal’s data sharing requirements. PLOS ONE does not permit references to inaccessible data.”

Answer: Our previous statement “Data not Shown” is corrected properly in the revised manuscript.

　　Response of the comments of Reviewer 1:

　　Major comments:

“The authors need to have every details of the reagents and chemicals used in each experiment in the manuscripts. …….. Moreover, the need to include the information on the respective primary antibody dilution in the Western blotting sub-section too”.

Answer: As per suggestion of the reviewer, details of the reagents and chemicals are provided in the revised manuscript along with catalogue numbers and vendors. Moreover, dilutions of the primary antibodies are mentioned in the revised text.

“In the biochemical assays sub-section of the materials and methods, The authors do not elaborate on the technique used to use different parameters like GSH, TBARS, protein content, CYP2E1, GPx, NOX”

Answer: As per suggestion of the reviewer, we elaborate all the biochemical techniques used in the revised text. 

The authors have performed the Sircol Collagen assay and presented in the result. However, there is no mention of the assay technique in the materials and methods section”.

Answer: Sircol Collagen assay technique is mentioned in the materials and methods section in the revised text.

Minor comments:

“Some of the paragraphs or portions of paragraphs are underlined within the manuscript. Is it done deliberately, what does it mean? I will suggest that the authors be consistent in their manuscript”.

Answer: Necessary changes are made in the revised manuscript as per suggestion of the reviewer.

The authors indicate in the statistical analysis that all the experiments have were replicated at least five times. None of the individual figures possess any information on the number of replicates used for each experiment. I will suggest that the authors provide the exact number of experimental replicated used to prepare the quantification graphs in each figure”.

Answer: Number of times the experiments were conducted is inserted within the figure legends.

　　Response of the comments of Reviewer 2:

Question 1: “In figure 1D, the expression of NOX was detected by western blot. What are NOX1 (T) and NOX1 (C)? The results and figure legend didn’t explain these items.

The results need to show the expression of NOX in untreated cells. After INH treatment, there is not the obvious increase in NOX1 expression between 24 and 72 h. But due to lack of results in the untreated cells, I didn’t know the change in NOX1 expression after INH treatment compared to untreated control”.

Answer: In figure 1D, NOX1 (C) and NOX1 (T) are explained in the figure legends. NOX1 (C) is untreated control and NOX1 (T) is INH treated cells. After treatment of INH, increase of NOX1 expression between 24 and 72 hours exhibited no change. There is no difference in expression of NOX1 between control [NOX1 (C)] and INH treated cells [NOX1 (T)].

Question 2: In Figure 4A, the results showed the CYP2E1 activity was increased after INH, PY or INH+PY treatment. Could the authors also show the results using western blot about the expression of CYP2E1 after INH, PY or INH+PY treatment?

Answer: As per suggestion of the reviewer, expression of CYP2E1 by western blot after INH, PY and INH+PY treatment is provided in the text and in figure 4D. 

Question 3: CYP2E1 is expressed mainly in hepatocytes and generate ROS. Accumulation of ROS serves as a driving force for HSC. ……………the authors used pyrazole (PY) pre-treated HSC to increase the expression of CYP2E1. … affect the HSC activation in vitro culture assay.

Answer: Since it is a short term study, to enhance CYP2E1 activity in LX2 cells, we have used PY to induce CYP2E1.

Question 4: The results showed INH or PY treatment can induce HSC activation. Upon activation HSCs express a combination of MMPs and TIMPs to remodel the ECM and facilitate the progression of liver fibrosis. How about the proliferation of HSC and MMP expression after INH or INH+PY treatment? 

Answer: As per suggestion of the reviewer, proliferation of HSC cells after INH or PY+INH treatment is inserted within the revised text. Further, results of MMP 2 amd MMP 9 are also provided in Table 2 as well as in the text.

Question 5: In figure 2C, the results showed pretreatment of LX2 cells with anti-oxidant or NOX inhibitor can reduce the intracellular ROS formation in response to INH treatment. How about the effect of CYP2E1 inhibitor in the same experiment?

Since the authors demonstrated that INH induced CYP2E1 expression can activate HSC, whether the CYP2E1 inhibitor can reverse the effect in response to INH treatment?

Answer: ROS generation in presence of CYP2E1 inhibitor was significantly reduced which is provided in the text and figure 2C.

Activation of LX2 cells in presence of CYP2E1 inhibitor was markedly reduced which is provided in the text and figure 3C.

---

## [Decision Letter · Decision Letter 1]

4 Jun 2020

PONE-D-19-35497R1

Expression of type I collagen in response to Isoniazid exposure is indirect and is facilitated by collateral induction of Cytochrome P450 2E1: An in-vitro study

PLOS ONE

Dear Dr. Chowdhury,

Thank you for submitting your manuscript to PLOS ONE. After careful consideration, we feel that it has merit but does not fully meet PLOS ONE’s publication criteria as it currently stands. Therefore, we invite you to submit a revised version of the manuscript that addresses the points raised during the review process.

One of the reviewer raised a minor comments which need to be addressed. 

We look forward to receiving your revised manuscript.

Kind regards,

Partha Mukhopadhyay, Ph.D.

Academic Editor

PLOS ONE

Reviewers' comments:

Reviewer's Responses to Questions

**Comments to the Author**

1. If the authors have adequately addressed your comments raised in a previous round of review and you feel that this manuscript is now acceptable for publication, you may indicate that here to bypass the “Comments to the Author” section, enter your conflict of interest statement in the “Confidential to Editor” section, and submit your "Accept" recommendation.

Reviewer #2: (No Response)

2. Is the manuscript technically sound, and do the data support the conclusions?

Reviewer #2: Yes

3. Has the statistical analysis been performed appropriately and rigorously? 

Reviewer #2: Yes

4. Have the authors made all data underlying the findings in their manuscript fully available?

Reviewer #2: Yes

5. Is the manuscript presented in an intelligible fashion and written in standard English?

Reviewer #2: Yes

6. Review Comments to the Author

Reviewer #2: In Fig1, Fig2A and Fig3B and 3D, the authors should present the data at 0 hours before INH treatment.

The author mentioned that INH treatment only induce low level CYP2E1 expression in Line 448. The authors used PY as a chemical inducer to induce the CYP2E1 expression. The expression of CYP2E1 induced by PY has no obvious difference compared to the INH treatment in Fig 4. Why the PY as a chemical inducer of CYP2E1 alone could not significantly increase the CYP2E1 level compared to the INH treatment? Only INH+PY treatment can significantly increase the CYP2E1 level compare to the INH treatment alone?

7. PLOS authors have the option to publish the peer review history of their article (what does this mean?). If published, this will include your full peer review and any attached files.

Reviewer #2: No

---

## [Author Response · Author response to Decision Letter 1]

11 Jul 2020

Response to Reviewers

We would like to thank the academic editor and the reviewer for critical review of our manuscript and for their thoughtful comments and constructive suggestions, which help to improve further the quality of this manuscript. Point wise answers of the comments of the reviewer are follows: 

　　Response of the comments of Reviewer 2:

Question 1: “In Fig1, Fig2A and Fig3B and 3D, the authors should present the data at 0 hours before INH treatment”.

Answer: As per suggestion of the reviewer, we have included the data at 0 hours before INH treatment in Fig 1; Fig 2A and Fig 3B and 3D in our revised version of the manuscript. 

Question 2: “The author mentioned that INH treatment only induce low level CYP2E1 expression in Line 448. The authors used PY as a chemical inducer to induce the CYP2E1 expression. The expression of CYP2E1 induced by PY has no obvious difference compared to the INH treatment in Fig 4. Why the PY as a chemical inducer of CYP2E1 alone could not significantly increase the CYP2E1 level compared to the INH treatment? Only INH+PY treatment can significantly increase the CYP2E1 level compare to the INH treatment alone?”

Answer: In our study, we used a single dose of CYP2E1 inducer pyrazol (PY) at a lower dose to examine any synergistic effect of CYP2E1 expression in PY pre-treated LX2 cells, during INH treatment. Using a dose of 50 µM of PY alone, expression of CYP2E1 expression in LX2 cells was comparable to INH treatment. At this level of CYP2E1 expression in LX2 cells, the increased oxidative stress developed in the cells could not produce collagen 1 which is a major pathological extracellular matrix. Treatment of INH in PY pre-treated LX2 cells, produce synergistic effect on CYP2E1 induction in LX2 cells, resulting increased CYP2E1 mediated increased oxidative stress, activation of LX2 cells that can produce collagen 1.

---

## [Editor Report · Decision Letter 2]

20 Jul 2020

Expression of type I collagen in response to Isoniazid exposure is indirect and is facilitated by collateral induction of Cytochrome P450 2E1: An in-vitro study

PONE-D-19-35497R2

Dear Dr. Chowdhury,

We’re pleased to inform you that your manuscript has been judged scientifically suitable for publication and will be formally accepted for publication once it meets all outstanding technical requirements.Congratulation for the great work!

Kind regards,

Partha Mukhopadhyay, Ph.D.

Section Editor

PLOS ONE
---

## [Editor Report · Acceptance letter]

22 Jul 2020

PONE-D-19-35497R2 

Expression of type I collagen in response to Isoniazid exposure is indirect and is facilitated by collateral induction of Cytochrome P450 2E1: An in-vitro study 

Dear Dr. Chowdhury:

I'm pleased to inform you that your manuscript has been deemed suitable for publication in PLOS ONE. Congratulations! Your manuscript is now with our production department. 

Kind regards, 

on behalf of

Dr. Partha Mukhopadhyay 

Section Editor

PLOS ONE